# Establishment of Human Leukocyte Antigen-Mismatched Immune Responses after Transplantation of Human Liver Bud in Humanized Mouse Models

**DOI:** 10.3390/cells10020476

**Published:** 2021-02-23

**Authors:** Akihiro Mori, Soichiro Murata, Nao Tashiro, Tomomi Tadokoro, Satoshi Okamoto, Ryo Otsuka, Haruka Wada, Tomoki Murata, Takeshi Takahashi, Ken-ichiro Seino, Hideki Taniguchi

**Affiliations:** 1Department of Regenerative Medicine, Yokohama City University Graduate School of Medicine, 3-9, Fuku-ura, Kanazawa-ku, Yokohama, Kanagawa 236-0004, Japan; t176065b@yokohama-cu.ac.jp (A.M.); e173054a@yokohama-cu.ac.jp (N.T.); tadokoro@yokohama-cu.ac.jp (T.T.); sokamoto@yokohama-cu.ac.jp (S.O.); 2Division of Regenerative Medicine, University of Tokyo, 4-6-1, Shirokanedai, Minato-ku, Tokyo 108-8639, Japan; 3Institute for Genetic Medicine, Hokkaido University, Kita-15, Nishi-7, Sapporo, Hokkaido 060-0815, Japan; otsuka@igm.hokudai.ac.jp (R.O.); wada@igm.hokudai.ac.jp (H.W.); murata@igm.hokudai.ac.jp (T.M.); seino@igm.hokudai.ac.jp (K.-i.S.); 4Central Institute for Experimental Animals (CIEA), Kawasaki 210-0821, Japan; takeshi-takahashi@ciea.or.jp

**Keywords:** allograft rejection, human iPS cell, liver bud, humanized mouse

## Abstract

Humanized mouse models have contributed significantly to human immunology research. In transplant immunity, human immune cell responses to donor grafts have not been reproduced in a humanized animal model. To elicit human T-cell immune responses, we generated immune-compromised nonobese diabetic/Shi-scid, IL-2RγKO Jic (NOG) with a homozygous expression of human leukocyte antigen (HLA) class I heavy chain (NOG-HLA-A2Tg) mice. After the transplantation of HLA-A2 human hematopoietic stem cells into NOG-HLA-A2Tg, we succeeded in achieving alloimmune responses after the HLA-mismatched human-induced pluripotent stem cell (hiPSC)-derived liver-like tissue transplantation. This immune response was inhibited by administering tacrolimus. In this model, we reproduced allograft rejection after the human iPSC-derived liver-like tissue transplantation. Human tissue transplantation on the humanized mouse liver surface is a good model that can predict T-cell-mediated cellular rejection that may occur when organ transplantation is performed.

## 1. Introduction

Liver transplantation is a useful intervention for patients with liver disease; however, liver donor graft shortage is a severe problem worldwide [1,2]. For that reason, only a limited number of patients who are indicated for liver transplantation can actually undergo liver transplantation. As an alternative to liver transplantation, hepatocyte transplantation has been studied [2,3]; however, the target diseases of this treatment are limited. Moreover, the source of hepatocytes is limited; therefore, the liver graft shortage and liver transplantation problems remain unresolved. A new alternative therapy that can replace both liver and hepatocyte transplantations has been expected for a long time.

Recently, human-induced pluripotent stem cells (hiPSCs) have been studied for regenerative medicine [4]. For example, studies on hiPSCs of the retina [5], myocardium [6], and neuron [7] are in progress. However, only some of the cells are able to be differentiate from hiPSCs, thus making it difficult to generate a whole organ from hiPSCs. Recently, an organoid study has made great strides to reconstruct a complicated organ like the structure in vitro by coculturing several cells [8,9,10]. This technology reconstructs not only normal tissues but, also, cancer tissues [8]. Previously, we reported a “liver bud”, which consists of hiPSC-derived hepatic endoderm cells, endothelial-like cells, and mesenchymal-like cells [11,12,13,14]. The liver bud mimics ED10.5 murine fetal livers [11]. These cells work together. The advantage of the liver bud is introducing vasculogenesis in vitro [14]. We also reported that liver bud transplantation improved the survival rate of mice models with acute liver disease [13].

The problem of a study using human-derived cells or tissue transplantation is the severe immune response, leading to xeno-rejection. For that reason, immunodeficient animals with T-cell, B-cell, and natural killer (NK)-cell depletions have been generated. Alternatively, these immunodeficient animals do not reproduce an allograft rejection after tissue and organ transplantations. Immune rejection is important in transplantation research. Generally, murine skin transplantation is often used for allograft transplantations. If the major histocompatibility complex (MHC) is nonconforming, the donor skin graft will be rejected [15]. In nonhuman primates, when there is an MHC mismatch, the allograft derived from iPSCs of nonhuman primates will be rejected [16]. In vitro experiments have demonstrated that the hiPSC-derived cells are attacked by allogenic T cells or NK cells and killed [17,18]. Using in vivo experiments of the human skin and islet, cardiac tissues have been transplanted in humanized mice [19,20]. The results showed allograft rejection. For the suppression of allograft rejection, clinical transplantation studies using hiPSCs should consider that homozygous human leukocyte antigen (HLA) cells or tissues that are matched with host HLA would be better than HLA mismatched grafts [21]. Recently, humanized mice with genetically expressed human HLA class I and HLA matched human immune cells have been developed. On using these mice, allogenic rejection is possible when human tissue, which has a different HLA class I, is used; however, it has not been reported yet [22]. Additionally, it is not clear if allograft rejection occurred because the hiPSCs-derived liver bud was transplanted to humanized mice.

The aim of this study was to reproduce allograft rejection by using our novel humanized mice and human iPSC-derived liver tissue. In this study, we succeeded in mimicking allograft rejection by transplanting allogenic hiPSCs, which are derived from liver buds, into humanized mice with HLA class I mismatches. Moreover, we succeeded in partially regulating this rejection using a calcineurin-inhibitor tacrolimus in the clinical transplantation method.

## 2. Materials and Methods

### 2.1. Animal Experiment

Nonobese diabetic/Shi-scid, IL-2RγKO Jic (NOG) mice were generated in the Central Institute for Experimental Animals (Kanagawa, Japan), as described previously [23]. The NOG mice were purchased from In-Vivo Science Inc. (Tokyo, Japan). All animal experiments were performed as per the ethical rules established by the Central Institute for Experiment and Yokohama City University’s animal experiment committee. Animal experiment protocols were approved by the animal experiment committee (approval number F-A-17-025, F-A-20-021).

### 2.2. NOG-HLA-A2Tg Generation

NOG-HLA-A2 transgenic (Tg) mice, formerly (NOD.Cg-Prkdcscid Il2rgtm1Sug Tg(HLA-A*0201/H2-Kb)A0201/Jic), were established as follows. HLA-A2 Tg mice on the C57/BL6 (B6) background were introduced from Taconic Biosciences (Albany, NY, USA) and used for backcrossing upon approval [24]. After more than seven times of backcross mating to the NOG, the replacement of the genetic background from B6 to NOD was confirmed by microsatellite markers (Appendix A).

### 2.3. Transplant Human Hematopoietic Stem Cell (HSC) to NOG-HLA-A2Tg

HSC transplantation was performed as described previously [23]. Six-week-old male or female NOG-HLA-A2Tg were irradiated with 160 cGy of X-rays (MBR-1520R-4, Hitachi, Hitachi, Japan), and the umbilical cord blood CD34^+^ cells (5 × 10^4^ cells, StemExpress, Lot# 1803310360, 1808220210, 1809270368, Folsom, CA, USA) were transplanted intravenously the next day. To analyze the human lymphocytes in mice reconstituted with the human immune system, a multicolor flow cytometric analysis was performed using a fluorescence-activated cell sorter (FACS) Fortessa (BD Biosciences, Franklin Lakes, New Jersey, USA). The peripheral blood (PB) was periodically collected from the retro-orbital venous plexus using capillary pipettes with sodium heparinization (Paul Marienfeld GmbH & Co.KG, Lauda-Königshofen, Germany) under anesthesia with isoflurane every four weeks. PB was also assessed using a blood analyzer (Microsemi LC-662, Horiba, Kyoto, Japan) to enumerate the total white blood cells (WBCs) and measure the hematocrit values (HCT). Red blood cells were lysed using an ammonium–chloride–potassium (ACK) solution (150-mM NH4Cl, 10-mM KHCO3, and 1-mM EDTA-Na2), and the mononuclear cells (MNCs) were stained with antibodies for flow cytometry. hCD45 chimerism was calculated using hCD45^+^ cells relative to the total CD45^+^ cells, which included the hCD45^+^ and mCD45^+^ cell populations.

### 2.4. Human-Induced Pluripotent Stem Cell Culture and Liver Bud Generation

The hiPSCs (Ff-I01s04) were maintained on Laminin 511 E8 fragment-coated (iMatrix-511, kindly provided by Nippi, Incorporated, Tokyo, Japan) dishes in StemFit AK02N (Ajinomoto, Tokyo, Japan). The hiPSCs were differentiated into Hepatic endoderm (HE), endothelial (EC), and mesenchymal cells (MC), as described previously [25].

The generation of hiPSC-derived liver buds was carried out as described previously [13]. We collected HE, EC, and MC and seeded them on Elplasia six-well plates (Corning, Corning, NY, USA). The seeded cell number for HE was 2.5 × 10^6^ cells, for EC, 1 × 10^6^ cells, and, for MC, 1 × 10^6^ cells per well. The culture medium used was the same as reported previously [25]. Y-27632 (FUJIFILM Wako Pure Chemical Corporation, Osaka, Japan) was added on day one. On day two, small liver buds were collected and reseeded on a cell culture insert (Corning, Corning, NY, USA). The medium was changed every other day.

### 2.5. Immuofluorescence

The presence of differentiating markers for HE, EC, and MC (HNF4a, CD31, and CD166, respectively) were checked by immunofluorescence. Briefly, the cells were fixed using 4% paraformaldehyde (PFA) and washed with 0.3% Triton X-100 in tris buffered saline (TBS). Then, the samples were blocked with protein serum-free block (Agilent Technologies Ltd., Santa Clara, CA, USA) and incubated with primary antibody (Table 1) at 4 °C overnight. The next day, the samples were washed three times with 0.3% Triton X-100 in TBS, and a secondary antibody was added at room temperature (RT) for one hour. Finally, after washing, the nuclei were stained with 4′,6-diamidino-2-phenylindole (DAPI) solution at RT for 10 min.

### 2.6. Tacrolimus Administrating

The group with tacrolimus was generated by administering 1-mg/kg tacrolimus (Astellas Pharma Inc., Tokyo, Japan) every day via intraperitoneal injection (i.p.) to the NOG-HLA-A2Tg mice, which were transplanted with HSC five days before transplantation.

### 2.7. Mouse Transplantation Procedure

The mice were transplanted with liver buds 14–25 weeks after HSC transplantation. The mice transplantation procedure was described previously [26]. The mice were anesthetized by isoflurane and the opened peritoneum. The mesothelium of the left liver lobe was peeled off, and a hiPSC-derived liver bud was transplanted onto the peeled-off portion. The transplanted cell number was 2 × 10^6^ cells in HE equivalents. After the transplantation, the transplant was covered with the middle lobe of the liver and the closed peritoneum.

### 2.8. Human Albumin Concentration in Mice Blood

Quantification of human albumin (ALB) was carried out as described previously [25]. All blood samples of the mice were collected before transplantation and then collected every week after transplantation. Blood samples were centrifuged by 4000 rpm, 20 min at 4 °C, and the plasma was collected. The collected plasma was diluted, and the human albumin concentrations were assessed by enzyme-linked immunosorbent assay (Bethyl Laboratories, Montgomery, TX, USA), according to the manufacturer’s instructions.

### 2.9. Histological Analysis

The transplanted graft was fixed with 10% formalin solution and was embedded in paraffin made into a paraffin block. The paraffin block was sliced at 5 µm. The samples were deparaffinized with xylene and a series of ethanol solutions and stained with hematoxylin and eosin. After washing with tap water, the samples were dehydrated with a series of ethanol solutions and cleared with xylene. Immunohistochemistry slides were deparaffinized, and the antigen was retrieved using a citric acid buffer (pH6.0). After blocking with the protein serum-free block, we added a primary antibody at 4 °C overnight (Table 1). The next day, the slides were washed three times with 0.3% TritonX-100 in TBS and were incubated with a secondary antibody at RT for one hour. After washing with 0.3% TritonX-100 in TBS, the nuclei were stained with 4′,6-diamidino-2-phenylindole (DAPI) solution. Images were obtained by Axio Imager M1 (Cael Zeiss, Oberkochen, Germany). 

### 2.10. Analysis

All imaging analyses were carried out using ImageJ 1.52a (National Institutes of Health, Bethesda, MD, USA). Statistical analysis was done by GraphPad Prism 8 (GraphPad Software, San Diego, CA, USA). We used two-way repeat measure ANOVA and Student’s *t*-test. All data were presented as the mean with standard error (SEM). *p* ≤ 0.05 was considered to be statistically significant.

## 3. Results

### 3.1. hiPSC-Derived Liver Bud Generation

We generated HE, EC, and MC from hiPSCs, as described previously [12,13] (Figure 1a). These results showed that HE cells were HNF4α (Hepatocyte nuclear factor 4 alpha)-positive, EC were CD31 (platelet endothelial cell adhesion molecule-1, PECAM-1)-positive, and MC were CD166 (activated leukocyte cell adhesion molecule, ALCAM)-positive (Figure 1a). Furthermore, we generated liver buds as per the protocols described previously [13] (Figure 1b). We checked for the generation of small liver buds in the Elplasia plate and reseeded these liver buds in the cell culture insert.

### 3.2. Generation of Humanized Mice

Humanized mice were generated by the transplantation of human HSCs in NOG-HLA-A2Tg mice. The human–lymphocyte ratio was investigated from mice PB using FACS (Figure 2a). Human T cells were observed in mice PB 12 weeks after transplantation (Figure 2b).

### 3.3. Allograft Rejection after hiPSC-Derived Liver Bud Transplantation to the NOG-HLA-A2Tg Mice

hiPSC-derived liver buds were transplanted to the surface of the murine liver as per the protocols described previously [26] (Figure 3a). The transplanted tissues were attached in all groups, as observed one and four weeks after transplantation. Hematoxylin and eosin staining showed the presence of HE-derived hepatocyte or bile duct clusters in the transplanted grafts (Figure 3b). In the NOG-HLA-A2Tg HSC-transplanted group (HSC(+)), more inflammatory cells were observed around the transplanted graft compared to the HSC(^−^) group (Figure 3b). Immunohistochemistry was conducted to determine whether the immune cells were derived from humans or mice (Figure 3c). The inflammatory cells that invaded in NOG-HLA-A2Tg HSC(+) were human CD45^+^ (Figure 3c). These human CD45^+^ cells increased one-to-four weeks after transplantation (Figure 3d). Next, we checked the characteristics of the hCD45 cells to determine whether these cells were T cells, monocytes, or NK cells. These cells were almost all T cells (CD3^+^ cells), with a few monocytes (CD14^+^ cells) (Figure 3e). One week after transplantation, an invasion of cytotoxic T cells (CD3^+^ and CD8^+^) was observed (Figure 3f,g). We also detected helper T cells (CD3^+^ and CD4^+^) around the transplanted graft (Figure 3f).

### 3.4. Suppression of Allograft Rejection by Administrating Tacrolimus

Tacrolimus was administrated five days before transplantation in the NOG-HLA-A2Tg HSC(+) group (Figure 4a). Four weeks after transplantation, inflammatory cell invasion decreased around the transplanted hiPSC liver bud in the tacrolimus(+) group compared to the HSC(+) tacrolimus(−) group (Figure 4b). Immunohistochemistry results showed less human CD45^+^ cell invasion in the tacrolimus(+) group compared to the tacrolimus(−) group (Figure 4c,d). The serum human albumin concentration in the HSC(+) tacrolimus(+) group was higher than the HSC(+) tacrolimus(−) group, as well as in the HSC(−) group, which indicates that the suppression of allograft rejection was partially achieved by administering tacrolimus (Figure 4e).

## 4. Discussion

In the present study, we revealed the acute allograft rejection mediated by cytotoxic T cells by transplanting allogenic hiPSC-derived liver buds into the newly developed humanized mice in an HLA-dependent manner.

NOG-HLA-A2Tg mice could reconstruct the human immune system without using human fetal tissues in the present study [18]. NOG-HLA-A2Tg mice could educate human T cells that have HLA-A2. These cytotoxic T cells attacked the allograft human tissue better than the currently used humanized mice, which are transplanted from nonspecific human HSC into normal immunodeficient mice without specific HLA expression [22]. In this study, we observed that the inflammatory cells invaded the transplanted human tissue in NOG-HLA-A2Tg mice with human HSC. These results revealed that the immunological rejection against the transplanted tissues occurred by host immune cells similar to human allograft rejection. We observed that more inflammatory cells invaded the human tissue in four weeks after transplantation compared to one week (Figure 3d). In the transplanted tissue, we detected cytotoxic and helper T cells four weeks after transplantation, which resembled acute cellular rejection in clinical settings (Figure 3f). Human tissue transplantation on the humanized mouse liver surface is a good model that can predict the T-cell-mediated cellular rejection that may occur when organ transplantation is performed.

By administering tacrolimus, human-derived inflammatory cells were inhibited from invading the transplanted tissue in this model (Figure 4b–d). The human serum albumin concentration was significantly higher in the tacrolimus group than in the group without tacrolimus and HSC (Figure 4e). These results suggest that allo-rejection was suppressed by the calcineurin-inhibitor tacrolimus in the humanized mice model.

The limitation of this study is that we did not induce humoral immunity using human B cells in the transplanted hiPSC-derived tissue. It is unclear whether T cells can be educated by HLA-restricted antigen-presenting cells. It is necessary to observe the humanized mice to maintain the allograft rejection ability for longer periods, because human lymphocytes in the humanized mice are not stable [22]. In the present study, we transplanted hiPSCs-derived liver buds to the surface of murine livers with normal functions. As our goal is to save patients with liver failure using hiPSCs-derived liver buds, it is necessary to consider human tissue transplantation in humanized immune mice with liver failure.

## 5. Conclusions

In conclusion, we succeeded in reproducing allograft rejection by transplanting human tissue to the liver surface of humanized immune mice. By transplanting HLA-mismatched human tissue on the organ surface of humanized immune mice, the severity of the alloimmune response can be predicted, and we can study the immune tolerance induction methods.

## Figures and Tables

**Figure 1 cells-10-00476-f001:**
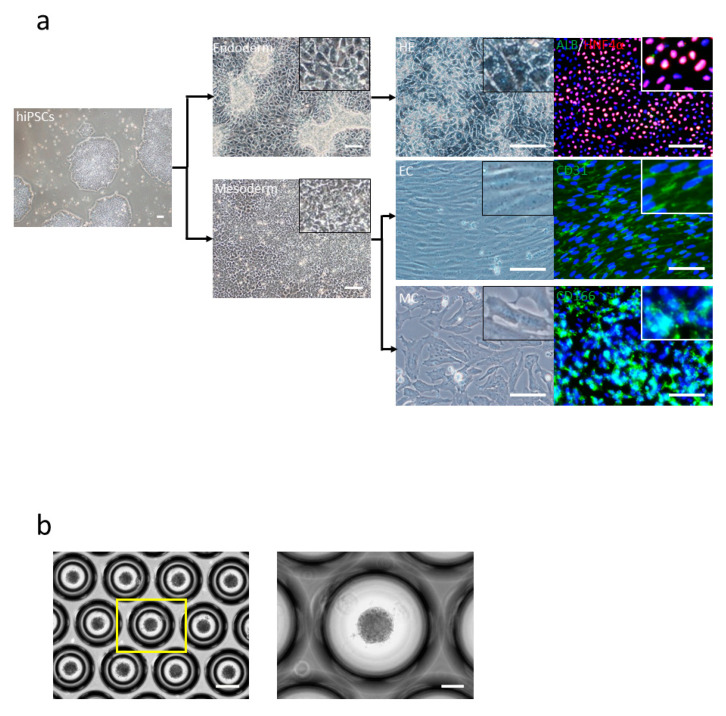
Human-induced pluripotent stem cell (hiPSC)-derived liver bud generation. (**a**) hiPSCs were differentiated to individual cells: hepatic endoderm (HE) (right-top) human albumin (ALB) (green) and HNF4α (red), endothelial cells (EC) (right-middle) CD31 (green), and mesenchymal cells (MC) (right-bottom) CD166 (green) and 4′,6-diamidino-2-phenylindole (DAPI) (blue). Scale bar: 100 µm. (**b**) Day 1 morphology of the liver bud is seen. Scale bar: 100 µm.

**Figure 2 cells-10-00476-f002:**
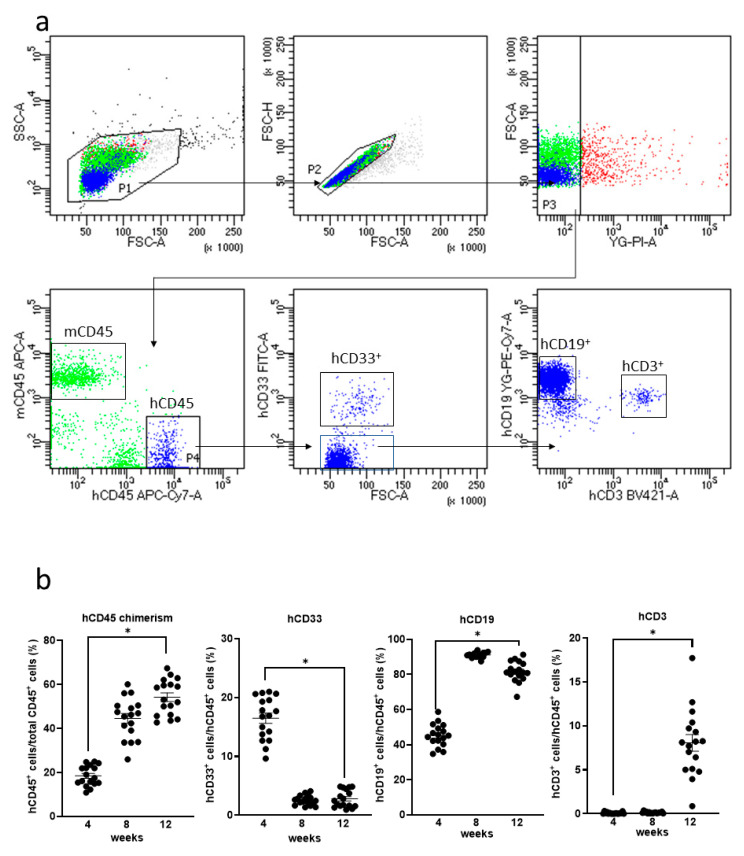
Human lymphocytes expression in the NOG-HLA-A2 Tg (transgenic) (NOD.Cg-Prkdcscid Il2rgtm1Sug Tg(HLA-A*0201/H2-Kb)A0201/Jic) mice peripheral blood. (**a**) Gating strategy of the human lymphocyte is shown. (**b**) Quantification of the human lymphocyte. hCD45 chimerism (left-top), hCD33^+^ cells (right-top), hCD19^+^ cells (left-bottom), and hCD3^+^ cells (right-bottom). Two-way repeated measure ANOVA with Tukey’s multiple comparisons tests were used. ns: no significant difference, * *p* < 0.0001, *n* = 17.

**Figure 3 cells-10-00476-f003:**
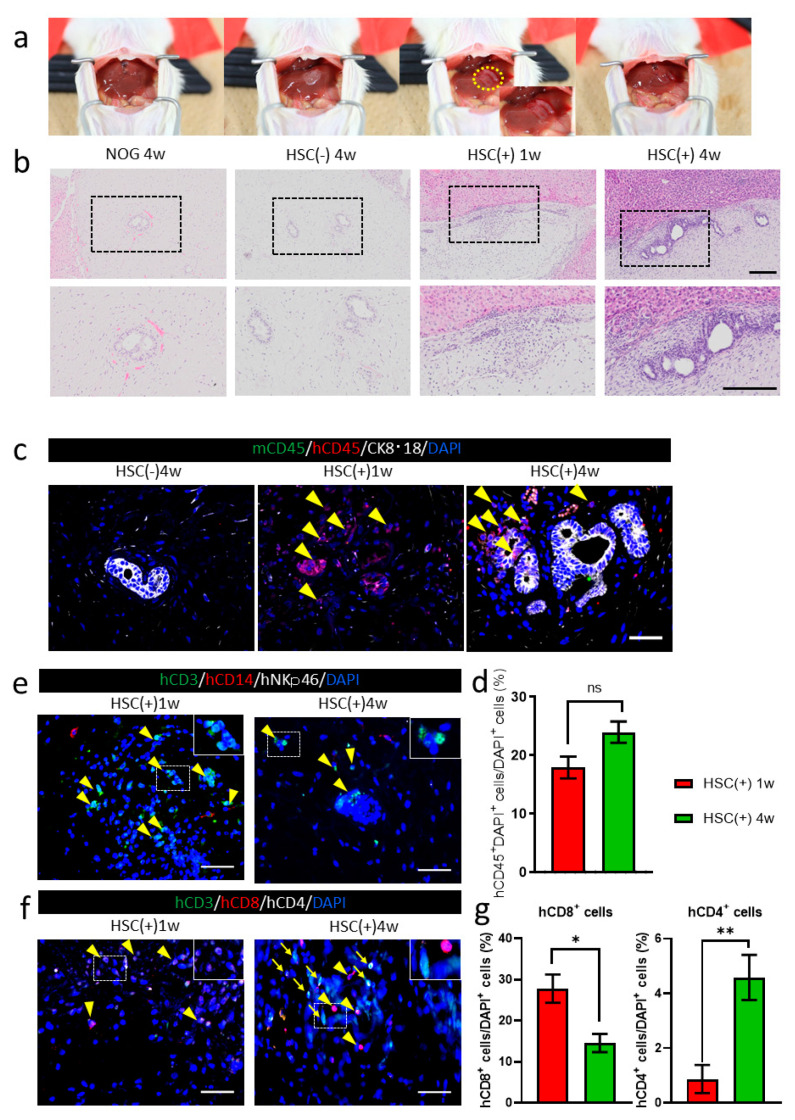
hiPSC-derived liver bud transplantations in NOG-HLA-A2Tg mice. (**a**) The image depicts the transplantation procedure. The yellow circle is the position where the liver bud was transplanted. (**b**) Hematoxylin and eosin staining. Top row shows low magnification, and bottom row depicts a zoomed-in image of the black frame border. Scale bar: 200 µm. (**c**) Immunohistochemistry for human lymphocyte detection: mCD45 (green), hCD45 (red), CK8·18 (white), and DAPI (blue). Arrowheads represent hCD45^+^ cells. Scale bar: 50 µm. (**d**) Count of the hCD45^+^ cells ratio. Error bar: SEM, Mann–Whitney *U* test, ns: no significance, HSC(+) 1w *n* = 18, and HSC(^+^) 4w *n* = 31. (**e**) Immunohistochemistry for T-cell detection is shown: hCD3 (green), hCD14 (red), hNKp46 (white), and DAPI (blue). Arrowheads show hCD3^+^ cells. Scale bar: 50 µm. (**f**) Immunohistochemistry for T-cell variation detection: hCD3 (green), hCD8 (red), hCD4 (white), and DAPI (blue). Arrowheads demonstrate hCD3^+^hCD8^+^ cells, and arrows show hCD3^+^hCD4^+^ cells. Scale bar: 50 µm. (**g**) Count of the hCD8^+^ cell and hCD4^+^ cell ratio. Error bar: SEM, Mann–Whitney *U* test, ns, * *p* < 0.01 and ** *p* < 0.0001, HSC(+) 1w *n* = 15, and HSC(^+^) 4w *n* = 22.

**Figure 4 cells-10-00476-f004:**
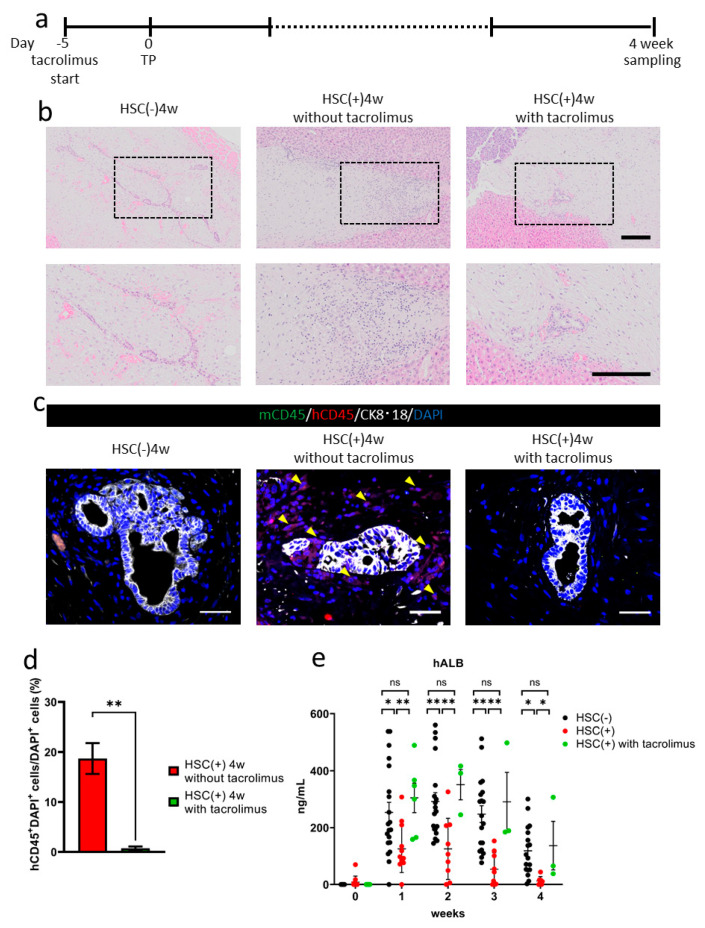
Tacrolimus administration to NOG-HLA-A2Tg. (**a**) The tacrolimus administrating schedule is shown. (**b**) Hematoxylin and eosin staining. Top row shows low magnification, and bottom row shows a zoomed-in image of the black frame border. Scale bar: 200 µm. (**c**,**d**) Immunohistochemistry. (**c**) Human lymphocyte detection: mCD45 (green), hCD45 (red), CK8·18 (white), and DAPI (blue). Arrowheads show hCD45^+^ cells. Scale bar: 50 µm. (**d**) Count of the hCD45^+^ cells ratio. Error bar: SEM, Mann–Whitney *U* test. * *p* < 0.05, HSC(+) 4w in the absence of tacrolimus, *n* = 12 and HSC(+) 4 weeks with tacrolimus, *n* = 17. (**e**) Human albumin (hALB) concentration in mouse blood. Two-way repeated measures ANOVA with Tukey’s multiple comparisons test, ns: no significant difference, * *p* < 0.05 and ** *p* < 0.001, HSC(−), *n* = 18–20, HSC(^+^), *n* = 7–12, and HSC(+) with tacrolimus, *n* = 3–7.

**Table 1 cells-10-00476-t001:** Antibodies used in the study.

Antibodies	Source	Product Number
APC/Cyanine7 anti-human CD45 Antibody	Biolegend, San Diego, CA, USA	304014
APC anti-mouse CD45 Antibody	Biolegend, San Diego, CA, USA	103112
PE anti-human CD3 Antibody	Biolegend, San Diego, CA, USA	300308
APC anti-human CD19 Antibody FITC anti-human CD33 Antibody Albumin Antibody	Biolegend, San Diego, CA, USA Biolegend, San Diego, CA, USA Novus, E Briarwood Ave, Centennial, CO, USA	363006 303304 NBP1-32458
Anti-CD3 antibody (ab828)	Abcam, Cambridge, UK	ab828
Human CD4 Antibody	R&D, Minneapolis, MN, USA	AF-379
Human CD8 alpha Antibody	R&D, Minneapolis, MN, USA	MAB3801
CD14 Monoclonal Antibody(5A3)	Invitrogen, Waltham, MA, USA	MA5-14773
Anti-Human CD31, Endothelial Cell	Dako, Santa Clara, CA, USA	M0823
Human CD45 Antibody	R&D, Minneapolis, MN, USA	MAB1430
Anti-CD45 antibody [I3/2.3]	Abcam, Cambridge, UK	ab25386
Anti-CD166 antibody [ERP2759(2)] ab109215	Abcam, Cambridge, UK	ab109215
HumanNKp46/NCR1 Antibody	R&D, Minneapolis, MN, USA	MAB1850
Pab to Keratins K8/K18	Progen, Heidelberg, Germany	GP11
HNF4A Monoclonal Antibody	Thermo Fisher, Waltham, MA, USA	MA1-199
Donkey anti-Rat IgG (H + L) Highly Cross-Adsorbed Secondary Antibody, Alexa Fluor 488	Invitrogen, Waltham, MA, USA	A-21208
Donkey anti-Rabbit IgG (H + L) Highly Cross-Adsorbed Secondary Antibody, Alexa Fluor 488	Invitrogen, Waltham, MA, USA	A-21206
Goat anti-Mouse IgG2a Cross-Adsorbed Secondary Antibody, Alexa Fluor 488	Invitrogen, Waltham, MA, USA	A-21131
Donkey anti-Mouse IgG (H + L) Highly Cross-Adsorbed Secondary Antibody, Alexa Fluor 555	Invitrogen, Waltham, MA, USA	A-31570
Goat anti-Mouse IgG1 Cross-Adsorbed Secondary Antibody, Alexa Fluor 555	Invitrogen, Waltham, MA, USA	A-21127
Donkey anti-Goat IgG (H + L) Cross-Adsorbed Secondary Antibody, Alexa Fluor 647	Invitrogen, Waltham, MA, USA	A-21447
Donkey anti-Rabbit IgG (H + L) Highly Cross-Adsorbed Secondary Antibody, Alexa Fluor 647	Invitrogen, Waltham, MA, USA	A-31573
Goat anti-Mouse IgG2b Cross-Adsorbed Secondary Antibody, Alexa Fluor 647	Invitrogen, Waltham, MA, USA	A-21242
Goat anti-Guinea Pig IgG (H + L) Highly Cross-Adsorbed Secondary Antibody, Alexa Fluor 647	Invitrogen, Waltham, MA, USA	A-21450

## Data Availability

The data presented in this study are available on request from the corresponding author.

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
