# Peer review of "Establishment of Human Leukocyte Antigen-Mismatched Immune Responses after Transplantation of Human Liver Bud in Humanized Mouse Models"

_cells, 2021, doi:10.3390/cells10020476_

Round 1

Reviewer 1 Report

In this manuscript, Akihiro Mori et al. present their work about an HLA-mismatched allograft rejection model in humanized mice. NOG mice with transgenic expression of HLA-A2 were generated by crossing HLA-A2 Tg mice with NOG mice. Human hematopoietic stem cells were transplanted into irradiated mice followed by allograft transplantation of hiPSCs-derived liver buds.

Strategies to model graft rejection in solid organ transplantation are still lacking. In this sense, this study is very valuable. However, the work needs to be improved, and the complete results have to be shown before considering for publication.

Major:

  1. Please describe better the HLA mismatch. Include the HLA typing of the transplanted HSC and the hiPSCs used in the live bud generation. A describing chart could help readers.
  2. The author should show their cytometric assay results used to analyze peripheral blood lymphocytes after human HSC transplantation. Also, bring the supplementary Figure 1 to the main text.
  3. For my understanding, what the authors call immunocytochemistry throughout the manuscript is immunofluorescence instead. Please correct where necessary.
  4. Figure 1. Image quality should improve. A better-quality magnification could be added.
  5. Figures 2e and 2f. It is a little hard to distinguish/interpret the labeled cells. The authors could show image magnification and quantify/plot the frequencies of CD3+, CD4+, and CD8+ in the two times (one week and four weeks after transplantation) they are showing.
  6. Supplementary Figure 1. Related to the previous point. Why was the hCD3+/hCD45+ population completely depleted at 4 weeks and 8 weeks after HSC transplantation? How was the CD45 chimerism and the peripheral lymphocyte profile just before the liver bud transplantation?

Minor:

  1. The author could show (maybe as a supplementary figure) the microsatellite screening that corroborates the genetic background's replacement from B6 to NOD.
  2. Section 2.4. In my opinion, the immunocytochemistry (immunofluorescence?) technique should be described in an independent section and not be embedded into the "Human-induced pluripotent stem cell culture and liver bud generation" Section.
  3. Section 2.7. Were blood samples centrifuged at 400 rpm? Should it be 4000 rpm?
  4. Include the methods for CD45 chimerism test
  5. Fig 1a (first panel) Should it be hiPSC instead of hiPS?
  6. Some sentences were unclear or hard to follow. Examples: "Only one type of cell is impossible to develop new therapy which can replace liver transplantation" or "Recently, an organoid study has made great strides to reconstruct a complicated organ like the structure in vitro." Please rephrase where needed.
  7. Overall, editing of English language and style are required.

Author Response

Reviewers' Comments to Author:

Reviewer 1

Major:

Please describe better the HLA mismatch. Include the HLA typing of the transplanted HSC and the hiPSCs used in the live bud generation. A describing chart could help readers.

→Thank you for your important comment. HLA-A haplotype of hiPSCs (Ff-I01s04) is A24 and HLA haplotype of HSC and is HLA-A2. We inquired another haplotype to the purchased company, the company told us another haplotype is unknown.

The author should show their cytometric assay results used to analyze peripheral blood lymphocytes after human HSC transplantation. Also, bring the supplementary Figure 1 to the main text.
→Thank you for your comments. According to the reviewer 1’s comment, we brought supplementary Figure 1 to the main Figure 2 and added explanation (page 6, line 176-178).

For my understanding, what the authors call immunocytochemistry throughout the manuscript is immunofluorescence instead. Please correct where necessary.

→Thank you for the comment. According to the reviewer 1’s comment, we changed the term immunocytochemistry to immunofluorescence and modified the explanation (page 3, line 118-126).

Figure 1. Image quality should improve. A better-quality magnification could be added.

→Thank you for the comment. According to the reviewer 1’s comment, we changed Figure1 magnification image for a better-quality magnification.

Figures 2e and 2f. It is a little hard to distinguish/interpret the labeled cells. The authors could show image magnification and quantify/plot the frequencies of CD3+, CD4+, and CD8+ in the two times (one week and four weeks after transplantation) they are showing.

→Thank you for your valuable comments. According to the reviewer 1’s comment, we added higher magnification image in Figure 3e and 3f and calculated the hCD45+ ratio in Figure 3d and hCD8+ ratio in Figure 3g. Also, previous Figure 2 was moved on to Figure 3.

Supplementary Figure 1. Related to the previous point. Why was the hCD3+/hCD45+ population completely depleted at 4 weeks and 8 weeks after HSC transplantation? How was the CD45 chimerism and the peripheral lymphocyte profile just before the liver bud transplantation?

→Thank you for the important comments. After HSCs were transplanted NOG-HLA-A2 mice, T cell development was necessary to wait for over 12 weeks. According to the reviewer’s comment, we added the calculation method of hCD45 chimerism (page 3, line 103-105).

 Minor:

The author could show (maybe as a supplementary figure) the microsatellite screening that corroborates the genetic background's replacement from B6 to NOD.

→Thank you for the comments. According to the reviewer 1’s comment we added the microsatellite screening to supplemental Figure1.

Section 2.4. In my opinion, the immunocytochemistry (immunofluorescence?) technique should be described in an independent section and not be embedded into the "Human-induced pluripotent stem cell culture and liver bud generation" Section.

→Thank you for the comments. In Section 2.5 we created the immunofluorescence explanation according to the valuable comments (page 3, line 118-126).

Section 2.7. Were blood samples centrifuged at 400 rpm? Should it be 4000 rpm?

→Thank you for the comment. We corrected it to 4,000 rpm in page 4, line 141.

Include the methods for CD45 chimerism test

→Thank you for the comment. We added the calculation method of hCD45 chimerism in page 3, line 103-105.

Fig 1a (first panel) Should it be hiPSC instead of hiPS?

→Thank you for the comments. We corrected it to hiPSC instead of hiPS in Fig 1a according to the comments.

Some sentences were unclear or hard to follow. Examples: "Only one type of cell is impossible to develop new therapy which can replace liver transplantation" or "Recently, an organoid study has made great strides to reconstruct a complicated organ like the structure in vitro." Please rephrase where needed.

→ Thank you for the important comments. According to the reviewer’s comment we rephrased the sentence (42-45).

Overall, editing of English language and style are required.

→ Thank you for the comment. We have asked to re-edit English again for the company.

Reviewer 2 Report

Thank you very much for giving me the opportunity to review the manuscript on the “Establishment of HLA mismatched immune responses after transplantation of human liver bud in humanized mouse models”.

Major critique

Unfortunately, there is a lack of normal distribution of continues data in the used student T-test. Thus, the authors may get a statistician involved or use the Mann-Whitney U-test as appropriate to cover the requirements for random samples.

Further, the number of humanized mouse models and samples are not clearly pointed out by the authors.

Most importantly, the authors shall pay attention to the innate immune system, which weakens their results. It should be taken into consideration that the proinflammatory signals arising before the activation of the adaptive immune system are considered as important factors of graft rejection.

Author Response

Reviewers' Comments to Author:

Reviewer 2

Unfortunately, there is a lack of normal distribution of continues data in the used student T-test. Thus, the authors may get a statistician involved or use the Mann-Whitney U-test as appropriate to cover the requirements for random samples.

→Thank you for the important comment. According to the reviewer 2’s comment, we modified the statistics to Mann-Whitney U-test (Fig 3 & Fig4 Figure and legend).

Further, the number of humanized mouse models and samples are not clearly pointed out by the authors.

→Thank you for the comment. According to the reviewer’s comment, we fixed figure legend to clarify the number of the animals and samples (Fig3&Fig4).

Most importantly, the authors shall pay attention to the innate immune system, which weakens their results. It should be taken into consideration that the proinflammatory signals arising before the activation of the adaptive immune system are considered as important factors of graft rejection.

→Thank you for the very important comment. In our experiment, we detected very few human CD14+ in NOG-HLA-A2Tg mice so the human HSC derived innate immune system is weakly  involved in this allo rejection. Also, we used tacrolimus and clearly recover hALB concentration in mice peripheral blood as same as normal NOG mice transplanted human LB. Considering these results, we think innate immune system has limited effect and T cell effect is important to reject transplanted graft in this system.

Reviewer 3 Report

  Establishment of Human Leukocyte Antigen-Mismatched Immune Responses after Transplantation of Human Liver Bud in Humanized Mouse Models by Akihiro Mori , Soichiro Murata * , Nao Tashiro , Tomomi Tadokoro , Satoshi Okamoto , Ryo Otsuka , Haruka Wada , Tomoki Murata , Takeshi Takahashi , Ken-ichiro Seino , Hideki Taniguchi is an interesting paper. I have only one comment I think that the aim of the study is unclear, I can finally read it at the end of the discussion "our goal is to safe patients with liver Failure using hiPSCs I think that the paper must be rewritten with this aim in mind The authors want to develop a model of transplantation of hiPSC and study rejection this transplantation will induce and how to act on it

Author Response

Reviewers' Comments to Author:

Review3

Establishment of Human Leukocyte Antigen-Mismatched Immune Responses after Transplantation of Human Liver Bud in Humanized Mouse Models by Akihiro Mori , Soichiro Murata * , Nao Tashiro , Tomomi Tadokoro , Satoshi Okamoto , Ryo Otsuka , Haruka Wada , Tomoki Murata , Takeshi Takahashi , Ken-ichiro Seino , Hideki Taniguchi is an interesting paper. I have only one comment I think that the aim of the study is unclear, I can finally read it at the end of the discussion "our goal is to safe patients with liver Failure using hiPSCs I think that the paper must be rewritten with this aim in mind The authors want to develop a model of transplantation of hiPSC and study rejection this transplantation will induce and how to act on it

→Thank you for the very important suggestion. According to the reviewer 3’s comment we will rewrite the manuscript to clarify the aim of this study (70-71).

Round 2

Reviewer 1 Report

The authors have addressed all my concerns and the manuscript has been significantly improved to warrant publication

Reviewer 2 Report

Thank you very much for giving me the opportunity again to review this manuscript on “Establishment of  HLA mismatched immune responses after transplantation of human liver bud in humanized mouse models”. Considering the revision of the manuscript and the replacement of the student T-test with the Mann-Whitney U-test, as recommended by our end, the statistics sounds technically correct. The article is now acceptable from my point of view, if the used methods are approved and accepted by an immunologist.